# Primary break-up and atomization characteristics of a nasal spray

**Kendra Shrestha**[1], **James Van Strien**[1], **Narinder Singh**[2,3], **Kiao Inthavong**[1]*

**1** Mechanical & Automotive Engineering, School of Engineering, RMIT University, Bundoora, Australia, **2** Sydney Medical School, Faculty of Medicine and Health, University of Sydney, Sydney, Australia, **3** Dept of Otolaryngology, Head & Neck Surgery, Westmead Hospital, Sydney, Australia

* kiao.inthavong@rmit.edu.au

**Data Availability Statement:** High speed imaging files are available from rmit.figshare.com database (doi:10.25439/rmt.12546860).

**Funding:** We gratefully acknowledge the financial support provided by Garnett Passe and Rodney

## Abstract

The primary objective of this research was to extract the essential information needed for setting atomization break up models, specifically, the Linear Instability Sheet Atomization (LISA) breakup model, and alternative hollow cone models. A secondary objective was to gain visualization and insight into the atomization break up mechanism caused by the effects of viscosity and surface tension on primary break-up, sheet disintegration, ligament and droplet formation. High speed imaging was used to capture the near-nozzle characteristics for water and drug formulations. This demonstrated more rapid atomization for lower viscosities. Image processing was used to analyze the near-nozzle spray characteristics during the primary break-up of the liquid sheet into ligament formation. Edges of the liquid sheet, spray break-up length, break-up radius, cone angle and dispersion angle were obtained. Spray characteristics pertinent for primary breakup modelling were determined from high speed imaging of multiple spray actuations. The results have established input data for computational modelling involving parametrical analysis of nasal drug delivery.

## Introduction

Nasal sprays are commonly used to treat conditions such as nasal congestion and allergic rhinitis. They are also used as an alternate route of administration for systemic therapy in place of intravenous and oral routes. However, their effectiveness is compromised since the atomized droplets are delivered at high velocities leading to excessive deposition within the anterior nasal cavity [1–4]. Since the atomization process leads to the formation of droplets, it is evident that the process has a central role in influencing drug delivery and deposition within the nasal cavity.

Spray atomization is challenging to study because it is a very rapid process that changes a liquid into a droplet phase with an accelerated flow from zero to 15-20 m/s [5, 6] within 100 microseconds after leaving the nozzle. During atomization, droplet formation occurs due to instabilities in the liquid sheet produced from competing forces between fluid inertia and surface tension (characterized by the Weber number and nozzle geometry). By varying both spray parameters and fluid parameters, Kooij et al. [7] found that the droplet-size distribution was

Williams Memorial Foundation Conjoint Grant 2019
(to KI and NS).

**Competing interests:** The authors have declared
that no competing interests exist.

dependent on the liquid surface tension, nozzle type, and flow rate. Shao et al. [8] investigated the formation of sheets, ligaments and droplets in swirling liquid using high fidelity simulation (Direct Numerical Simulations, DNS) and identified two phenomena that led to the formation of ligaments. However, these DNS simulations are computationally expensive, and sub-models such as the LISA (Linearized Instability Sheet Atomization, [9]) breakup, or spray cone models are used for quicker numerical solutions and design optimization studies. However, these approaches require boundary conditions and specific values based on physical spray characteristics in order to activate the models, which are difficult to find in the literature for nasal spray applications.

In particular, the atomization dispersion angle has not been reported for nasal sprays. The dispersion angle is the time-averaged liquid sheet fluctuation angle from the mean spray cone angle, and therefore describes the random dispersion that can occur due to the natural wave-like fluctuations that occur during the break down of a swirling liquid sheet. For engine sprays, the dispersion angle is generally set as 10˚ [10–12]. Fung et al. [13] tuned the dispersion angle to match with a Sauter mean droplet diameter distribution and found a dispersion angle of 3˚ provided good matching with experimental data. This study aims to determine a typical value from repeated nasal spray actuations.

Computational Fluid Dynamics (CFD) studies of nasal spray drug deposition include Kimbell et al. [4] who found that the deposition fractions of 20 and 50μm particles exceeded 90% in the anterior part of the nasal cavity; and that deposition efficiency increased with smaller particles when the nozzle was placed 1cm into the nostril. Inthavong et al. [14, 15] found that the swirl fraction produced in a nasal spray could increase its drug deposition efficacy since increasing swirl fraction reduces the particle linear velocity and thus reduces its inertia. Furthermore, it was found that hollow cone spray types resulted in higher deposition in the middle regions of the nasal cavity, over full cone spray types. Dong et al. [16] and Tong et al. [17] found that a spray axis aimed along the center line of the nasal valve improved spray efficacy compared with the upper or lower directions within the nozzle orientation adjustment plane. However, these studies used monodispersed particle distribution with initial constant velocities which differs to the atomizer primary breakup models (e.g., LISA break up model) that generate both particle size distribution, and velocity.

External spray characteristics include spray content uniformity, spray pattern and plume geometry uniformity, spray delivery repeatability and pump to pump reproducibility, which represent quality and reliability of the nasal spray device over the entire cycle of actuations from a spray bottle. Some of these characteristics were measured by Liu et al. [18], where droplet size distribution and spray plume geometry were obtained. Dayal et al. [19] evaluated different parameters, including actuation force and drug viscosity, with eight different nasal spray pumps and assessed their effect on droplet size distribution. A linear correlation was observed between increased viscosity and droplet size distribution. Cheng et al. [3] evaluated four different nasal pumps and found that spray angle and droplet size of the nasal spray were critical parameters that influenced deposition pattern in the nasal airway. Kundoor and Dalby [20] and Sosnowski et al. [21] demonstrated nasal spray deposition patterns in a human nose model using a color-based gel which revealed that lower viscosity formulations provided greater coverage than a higher viscosity formulation.

The primary objective of this study was to extract the essential information from spray atomization using high speed filming, that is needed for setting atomization break up models. Specifically, the parameters targeted are those required in the Linear Instability Sheet Atomization (LISA) breakup model. In addition, parameters describing primary atomization values for solid cone or hollow cone type models will also be determined. While extensive studies of spray atomization exist, these have covered high-pressure applications found in combustion

and industrial sprays [22, 23]. For nasal sprays that exhibit low-pressures there is a lack of information.

High speed imaging was used to gain insight into the primary breakup mechanism occurring in the region near the nozzle for a drug formulation from an over-the-counter nasal spray. Image processing techniques were used to analyze and obtain values of spray characteristics during the primary break-up stage. This was achieved by extracting edges of the liquid sheet, determining spray break-up length, break-up radius, cone angle and dispersion angle. A secondary objective was to visualise the liquid sheet break-up, and formation of ligaments to gain insight into the mechanisms involved in the atomization break up process. The influence of viscosity and surface tension on primary break-up, sheet disintegration, ligament formation and droplet break-up was evaluated by repeating the experiments and replacing the drug formulation with water. It is expected that the results will provide realistic values required in atomizer primary break-up models in CFD studies, and to gain insight into the atomization mechanisms for lower pressure applications such as nasal spray atomization.

## Materials and methods

### High speed imaging

A schematic of the experimental setup is shown in Fig 1a which contains an automated actuation system. The pneumatic actuator (model: SMC-CXSL10-10; ADI Inc., Hatfield, Pennsylvania) was located under the bottle and was connected to a two-way solenoid valve, controlled by a programmable logic control (PLC) unit (model: Allen Bradley 1760-L12BWB). The spray bottle was fixed at its base onto the actuator, to avoid lateral motion during actuation. Speed controllers (model: SMC-AS2002F-06; Allied Electronics, Inc., Fort Worth, Texas) were mounted on the pressure lines to control the flow rate and thus maintain the speed of squeeze and release of the spray bottle. During actuation the spray bottle moved up and down with the platform it was attached to, while the spray nozzle position remained fixed. This allowed images to be captured with a fixed reference point. The strength of actuation force was controlled by a compressed air line and the pressure passing through was monitored and controlled by a pressure regulator to produce 5Bar (72.5Psi). The PLC unit consisted of mechanical switches, a timer and a counter which controlled the timing, and numbers of on and off activations of the solenoids. The mechanical signal emitted from the PLC unit was converted to a digital signal by a Schmitt trigger and was sent to trigger the digital camera for image acquisition.

The actuation station was placed between a high-speed camera and a 1000W light source to produce shadowgraph images. An additional spotlight was used to allow shorter exposure times. Spray images were captured by a Phantom V210 digital high-speed camera with $1280 \times 800$ CMOS sensor. The frame rate was 10,000 per second (i.e. 100μs per frame) with a resolution of $512 \times 384$ pixels.

An 'over-the-counter' nasal spray bottle (Dimetapp 12 Hour Nasal Spray 20mL, GlaxoSmithKline, Ermington, NSW, Australia) from a drug store/ pharmacy was used and had a recommended 200 actuations. The active ingredient was oxymetazoline hydrochloride at 500 micrograms/mL, and it also contained benzalkonium chloride at 0.02% w/v as a preservative. While it was expected that the spray plume and geometry uniformity would be maintained over the course of the 200 actuations in a single bottle, we performed repeated measurements divided into stages of the bottle volume, e.g. fully filled, half-filled and partially filled after successive actuations. The measured length of the dip tube was 3cm and the diameter of the bottle was 2.9cm totaling the liquid volume to 20mL in a bottle. The volume of the liquid and corresponding length of the dip tube immersed in the liquid is provided in Table 1. Actuations for

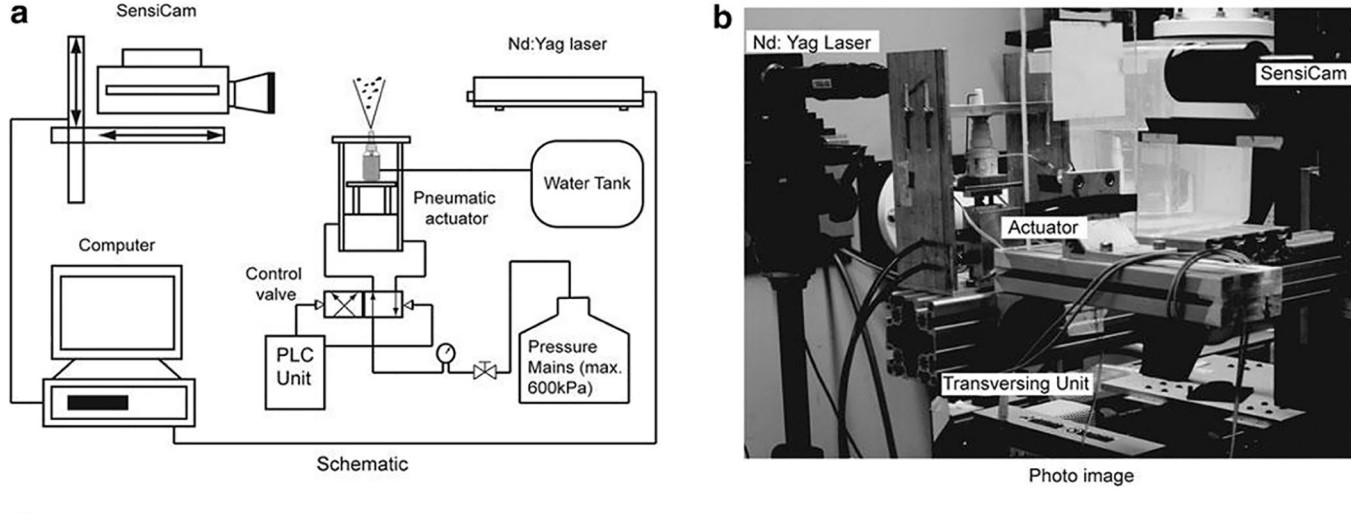

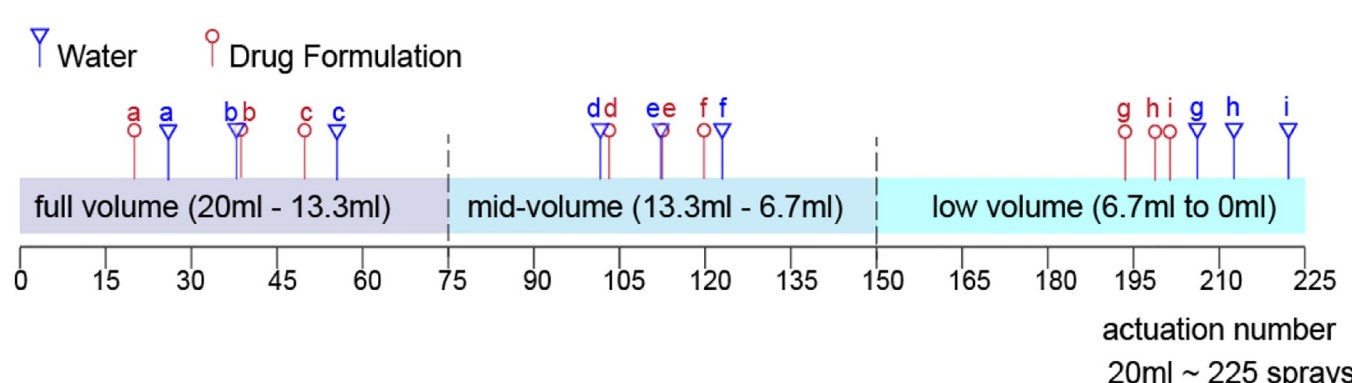

**Fig 1.** (a) Schematic of the experimental set-up for high-speed filming of spray atomization from a nasal device. (b) Photo image of the experimental set-up (c) Measurements taken at different liquid volumes of the bottle (full-, mid-, and low-volumes) during a single bottle use, labelled in red (for drug formulation) and blue (water).

priming and other actuations that were error prone (e.g. camera timing and syncing errors) were recorded and up to 225 actuations were achieved. The recorded actuations are shown in Fig 1c.

After the drug formulation was depleted, the bottle was refilled with distilled water, and the measurements were repeated for the water to serve as a reference case. Fluid properties of drug formulations are affected by the therapeutic needs of the solution, but can also be controlled with excipients such as buffers, solubilizers, preservatives, surfactants, bio-adhesive polymers and penetration enhancers leading to different viscosities and surface tension which influences

**Table 1. Measurements taken at different liquid volume of the bottle and corresponding length of the tube immersed representing full volume, mid-volume and low-volume bottle fills.**

| Bottle condition | Volume of liquid (mL) | Corresponding length of the submerged tube (cm) |
|---|---|---|
| Full-volume | 20-13.3 | 3-2 |
| Mid-volume | 13.3-6.7 | 2-1 |
| Low volume | 6.7-0 | 1-0 |

**Table 2. Liquid fluid properties of water compared with range of drug formulation properties.**

| Fluid Properties at 25˚C | Water | Nasal Sprays (range) | Nasal spray formulation used in current study[a] |
|---|---|---|---|
| Dynamic Viscosity (cP) | 0.89 | 655 – 3761[b] | 923 |
| Surface tension (N/m) | 72 | 30 – 44 | 40.6 |

[a] Properties of oxymetazoline HCl 0.05%

[b] Doughty et al. [24] reported Afrin with 3761 cP, and Zycam with 655 cP

the atomization process [19]. Table 2 shows fluid properties of water compared to drug formulation values reported in the literature.

## Image processing

Image processing was performed using MATLAB (MathWorks, Natick, Massachusetts) to obtain the near-nozzle spray characteristics of spray cone angle, dispersion angle, break-up radius and break-up length. The Canny edge detection algorithm [25] was used to determine the edge boundary by finding the greatest gradient of change of the intensity of pixels. The boundary was defined where a non-zero pixel value was found, while the background had a pixel value of zero (black). By converting the number of pixels with value of zero between the two edges of the spray cone, several transient spray parameters were obtained [6, 26–28]. The edge detector was applied to each image where of the high-speed recordings at a frame rate of 10,000 images per second.

# Results and discussion

## Early pre-stable phase of spray plume development

During a nasal spray actuation, the primary breakup can be categorized into three phases based on near-nozzle spray plume shape as: i) pre-stable or expanding phase, ii) fully developed stable phase and iii) collapsing phase. Fig 2 shows images of the spray plume development during the pre-stable phase occurring at time $t$ = 1.5ms to 4.0ms after the liquid first exits the nozzle. A comparison is made between the drug formulation and water. There is a clear difference in penetration length of the liquid sheet between the drug formulation and water as the liquid ejects from the nozzle with high momentum to overcome the surface tension. This was consistent for all other repeated actuations.

The boundary edges of the spray for different actuations (actuations labelled a,b,c in Fig 1c) were determined, representing the penetration length with time. Fig 3 shows the liquid sheet edges overlayed in colour at different time instances (1.5ms to 4ms with a 0.5ms interval), and in black lines, representing Canny edge detection of all images captured (30 images over 3ms) during the pre-stable phase of spray development. The images demonstrate a highly unstable and oscillating liquid sheet. Water, with a lower viscosity, produced a faster spray plume development (full cone within 4ms) compared to the drug formulation (full cone angle within 6ms). Therefore, transition to the fully developed stable stage of the spray was observed much earlier for water compared to the drug formulation.

## Stable phase of spray plume development

The ejected liquid forms a swirling sheet and is fully developed during the stable stage of the spray. Fig 4 shows images at the beginning of the fully-developed stable stage where it is clearly formed after time $t$ = 6ms. The liquid sheet swirls as a result of the tangential ports within the

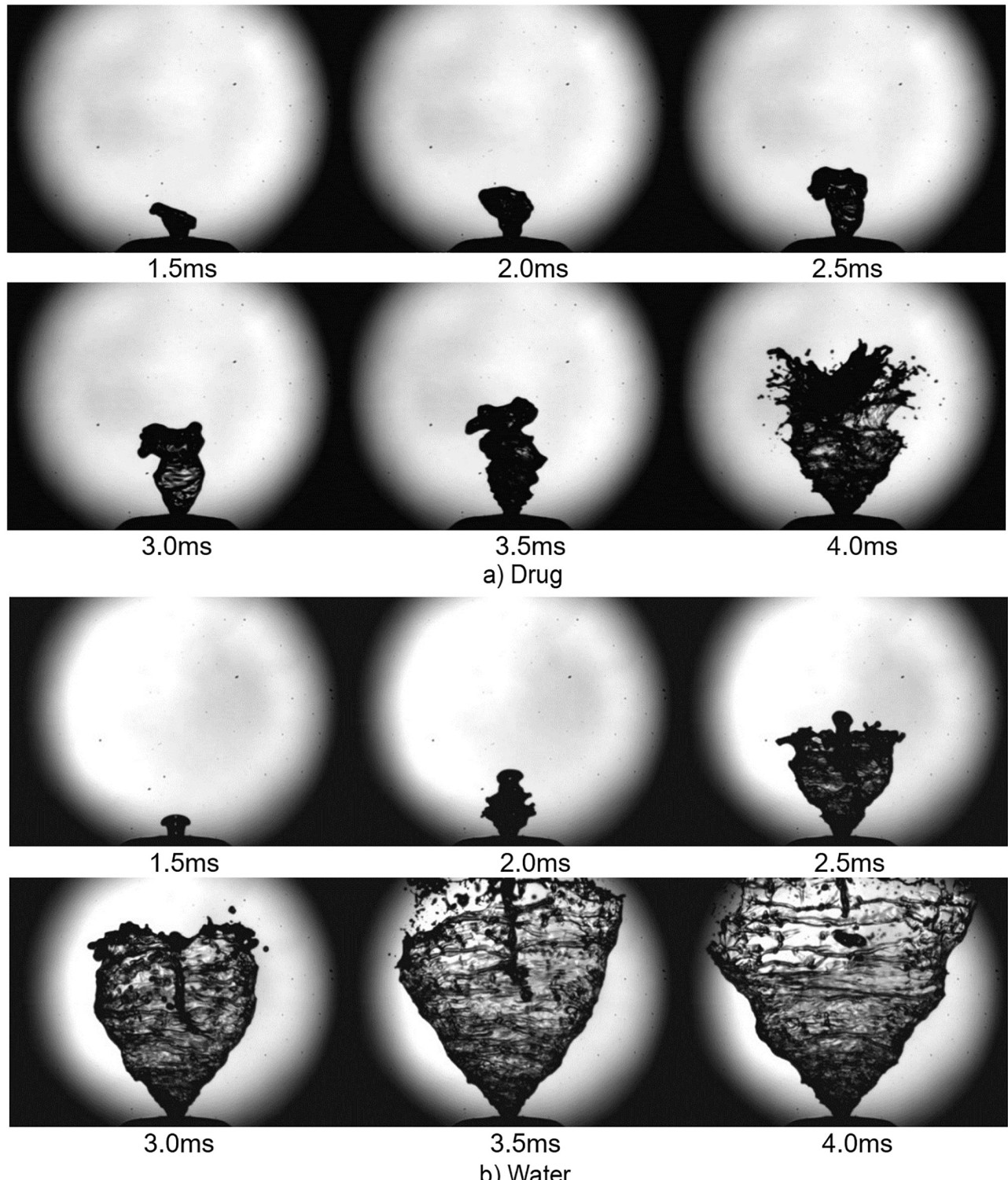

**Fig 2. Spray plume development during pre-stable phase.** Refer Fig 1(c) for corresponding measurement records evaluated in the figure with red and blue colored annotations for (a) drug and (b) water respectively.

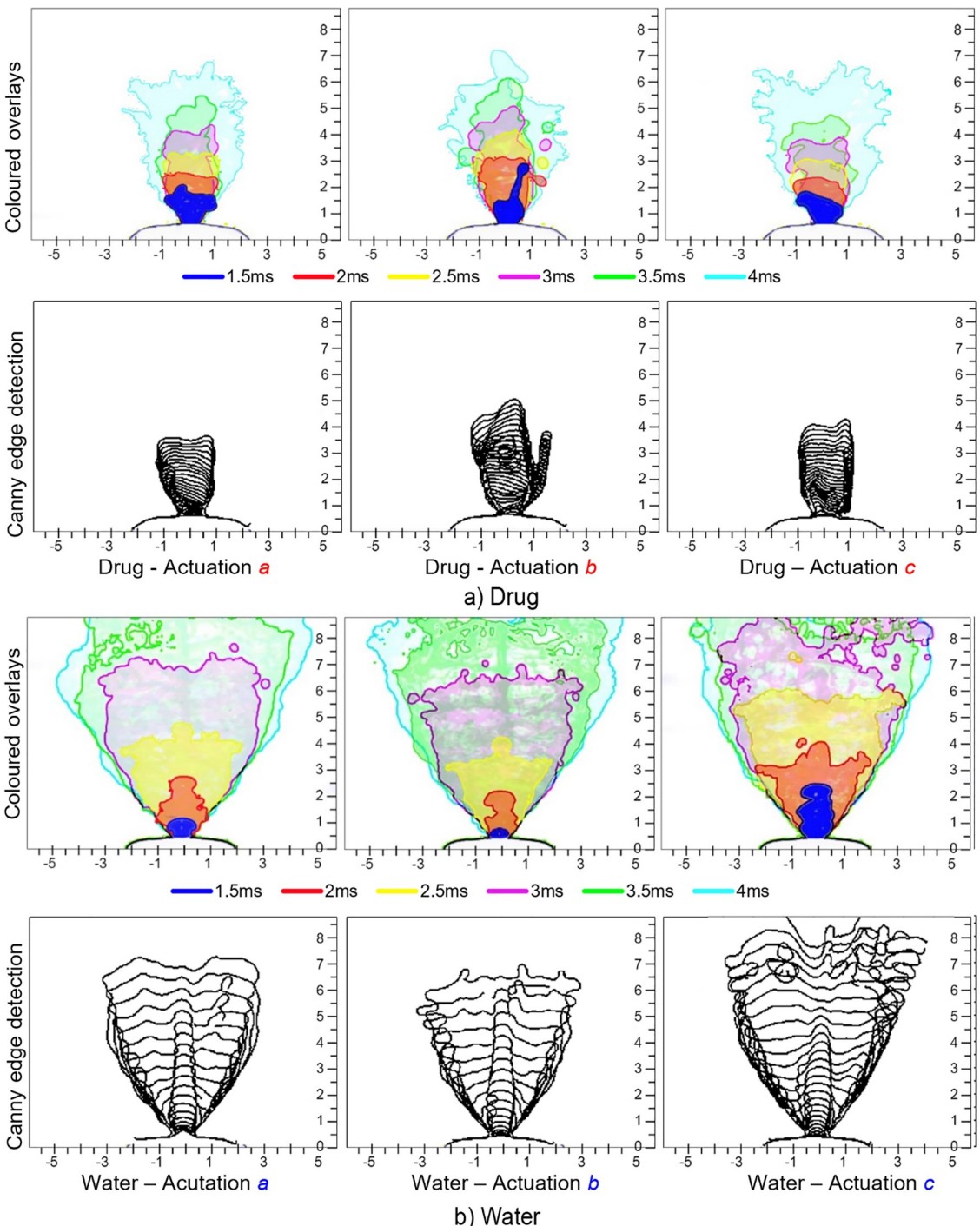

**Fig 3. Superimposed liquid sheet showing spray plume development from *t* = 1.5ms to 4ms; and Canny edge detection (threshold 0.7 and sigma 6) overlay of the liquid sheet from *t* = 0ms to 3ms (30 edge detected images), during pre-stable phase.** Spray actuations labelled a,b,c are referenced to Fig 1(c). (a) Drug formulation, and (b) Water.

pressure-swirl atomizer ejecting the liquid out of the nozzle with tangential velocity. Disturbances on the liquid sheet surface can be seen and are caused by the shear force from ambient air. As the spray expands, the disturbances grow in amplitude to overcome the liquid viscosity and surface tension, leading to liquid sheet breakup.

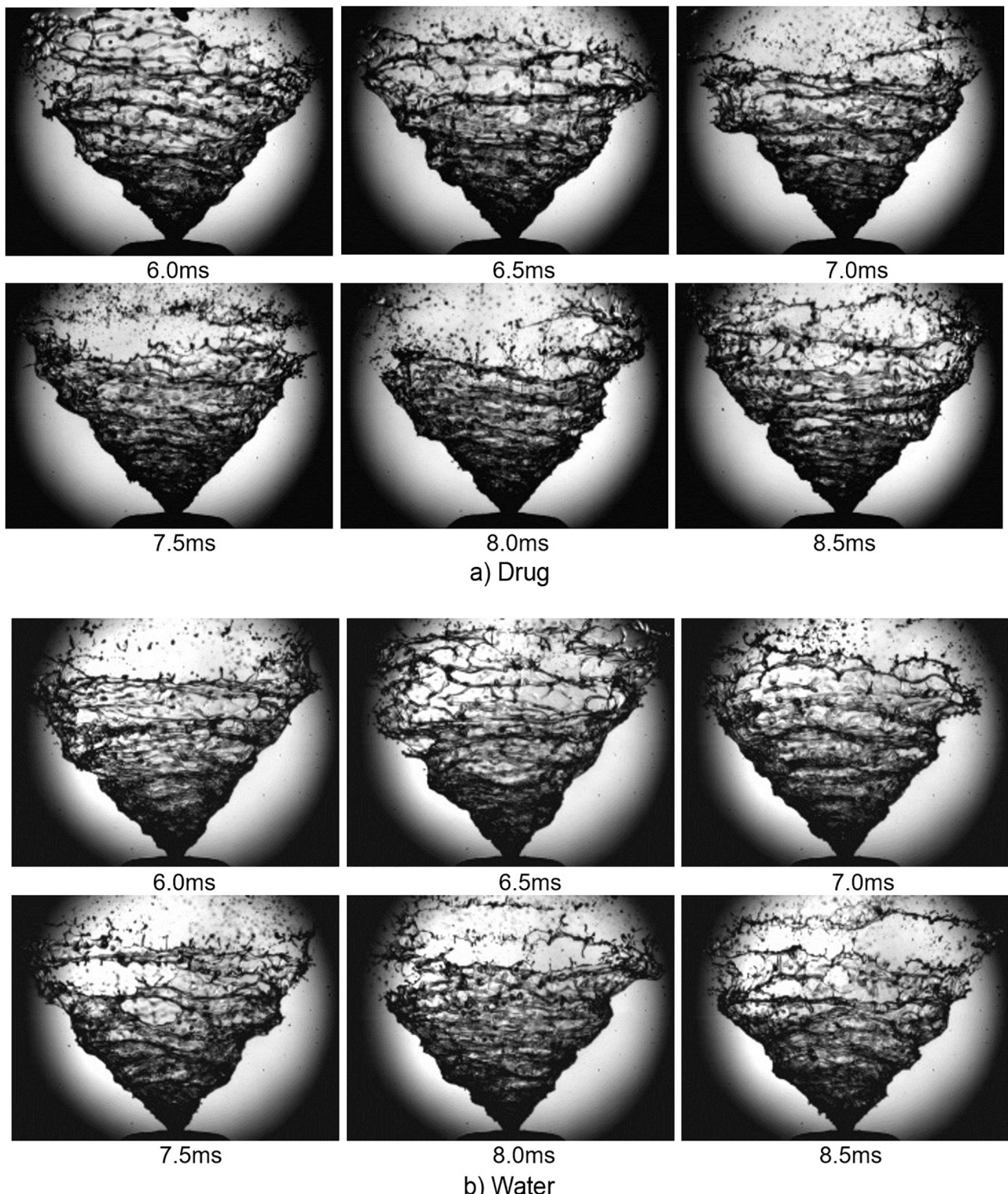

**Fig 4. Spray plume development during stable phase.** (a) Drug formulation for actuation a. (b) Water for actuation a.

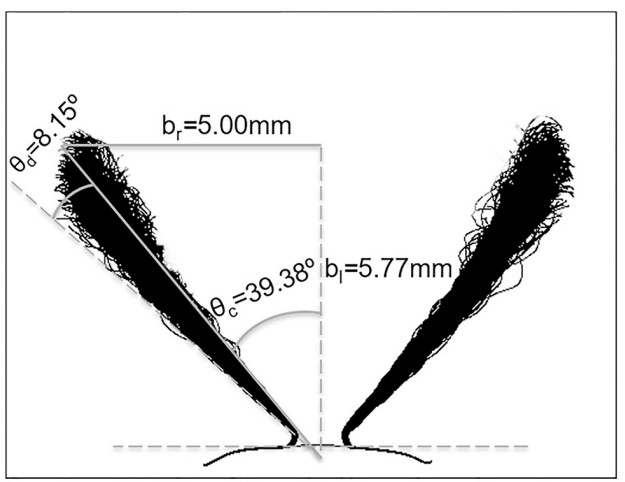
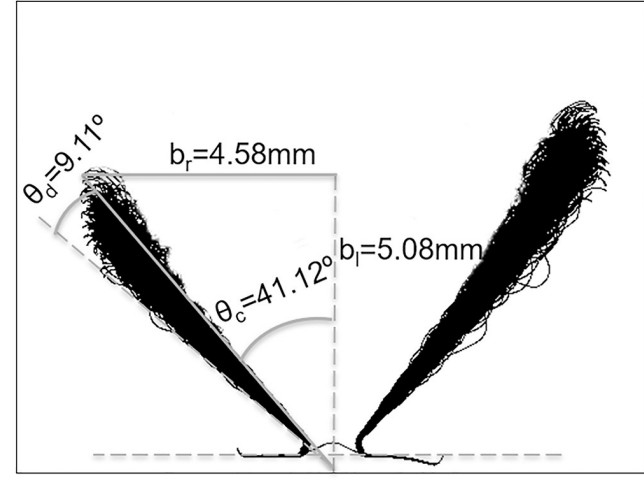

a) Drug - actuation a

b) Water - actuation a

**Fig 5. Canny edge overlay during stable phase of spray development (Canny edge detection threshold = 0.90 and sigma = 8).** In the figure $\theta_c$ is the spray half cone angle, $\theta_d$ is the dispersion angle, $b_r$ is the break-up radius and $b_l$ is the break-up length. Refer Fig 1c for corresponding measurement records evaluated in the figure (a) Drug formulation for actuation a. (b) Water for actuation a.

To determine the dispersion angle, the Canny edge detector was applied to extract the edge boundary from each image, and the results were then time-averaged. Fig 5 shows the time-averaged liquid sheet edge fluctuation (overlayed images) during the stable stage, demonstrating its temporal variation during time $t$ = 5ms to 45ms (400 images). The liquid sheet length where sheet breakup occurs was defined as the break-up length. At this distance thread-like liquid structure detaches from the continuous liquid sheet to form ligaments. Furthermore, the radius of the swirling hollow cone at the break-up length is an important parameter for CFD nasal drug delivery studies, since this is where initial droplet positions are located in CFD modelling and the necessity of computationally expensive fully-resolved primary break-up modelling can be avoided. The spray cone half angle for the specific nasal spray device was 39.38˚, dispersion angle was 8.15˚ and breakup length was 5mm for the drug formulation, while for water it was 41.12˚, 9.11˚, and 4.58mm, respectively. A comparison with data in the literature is summarised in Fig 6. These parameters provide a range of realistic values that

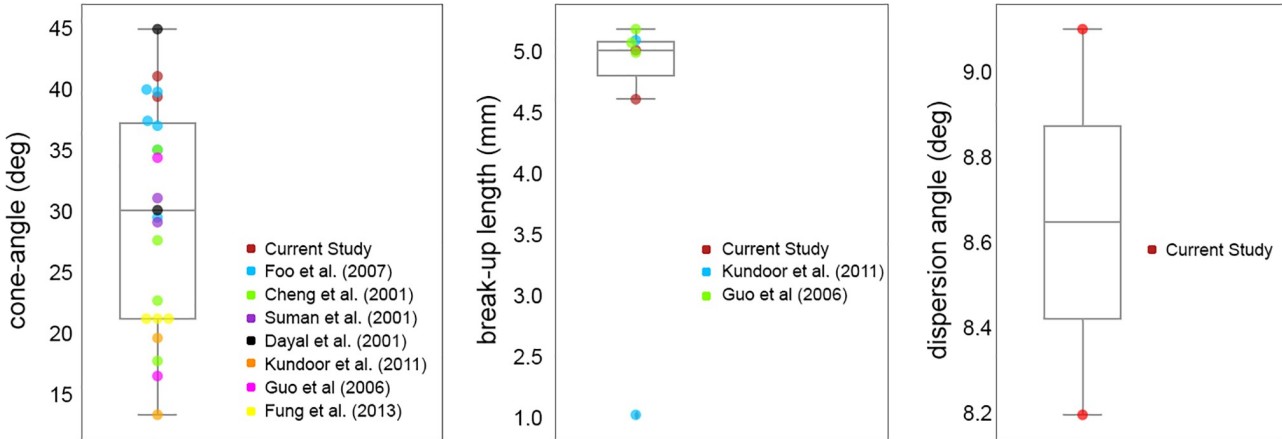

**Fig 6. Box plot summarising the reported data (as overlayed scatter points) in the literature [3, 19, 20, 29–31].** The statistical descriptions for each parameter arecone-angle: mean = 29.68˚, std = 9.15; break-up length: mean = 4.56mm, std = 1.13; dispersion angle = mean = 8.65˚; std = 0.64.

**Table 3. Summary of near-nozzle spray characteristics during fully developed spray at different bottle volumes.**

| Bottle Condition | | full volume | | | half volume | | | low volume | | |
|---|---|---|---|---|---|---|---|---|---|---|
| Drug | Actuation | a | b | c | d | e | f | g | h | i |
| | Half angle (deg) | 40.0 | 39.2 | 38.9 | 39.1 | 38.8 | 39.0 | 34.9 | 33.4 | 28.2 |
| | Break-up radius (mm) | 5.0 | 5.0 | 4.84 | 5.0 | 4.9 | 5.0 | 3.0 | 2.7 | 2.1 |
| | Break-up length (mm) | 5.8 | 5.9 | 5.8 | 5.9 | 5.9 | 6.0 | 4.1 | 4.0 | 3.4 |
| | Dispersion angle (deg) | 8.1 | 9.0 | 9.4 | 9.6 | 10.0 | 9.1 | 14.3 | 13.6 | 18.5 |
| Bottle Condition | | full volume | | | half volume | | | low volume | | |
| Water | Actuation | a | b | c | d | e | f | g | h | i |
| | Half angle (deg) | 41.1 | 41.2 | 41.7 | 41.6 | 41.1 | 41.8 | 40.5 | 39.9 | 35.3 |
| | Break-up radius (mm) | 4.6 | 4.8 | 4.9 | 4.7 | 4.6 | 4.8 | 4.5 | 4.1 | 3.8 |
| | Break-up length (mm) | 5.1 | 5.3 | 5.3 | 5.2 | 5.1 | 5.2 | 5.1 | 4.8 | 5.2 |
| | Dispersion angle (deg) | 9.1 | 8.4 | 8.6 | 7.9 | 9.2 | 8.6 | 9.2 | 8.9 | 12.0 |

form part of the conditions required to activate CFD atomization models including the Linear Instability Spray Atomization (LISA) model [9].

In general, an increase in actuation pressure reduces the liquid sheet break-up length (i.e. faster atomization), but an increase in liquid viscosity reverses this effect [32], since it holds the fluid together and resists disruption and instabilities. The higher viscosity in the drug formulation produced a longer break-up length and smaller dispersion angle (Fig 5). The lower viscosity of water allowed more intense instability (Kelvin-Helmholtz) and vigorous flapping motion (i.e. larger dispersion angle) leading to faster liquid sheet disintegration (i.e. shorter breakup length). Variation in near-nozzle spray characteristics at different stages of spray actuations taken at full, half and low bottle volumes, to ensure repeatability in the results, are summarized in Table 3, while the qualitative images are shown in Fig 7. The results suggest that there are insignificant changes in the spray parameters under fully-filled and half-filled bottle conditions. However, differences were observed in the late actuations which represent a low-filled bottle volume. Generally, the stable phase of the fully developed spray occurred for 40ms (between 5ms to 45ms) before the liquid sheet started to collapse into a spiral shaped liquid column. However, this duration was shortened when the bottle volume was depleted.

## Primary breakup visualization and analysis

Fig 8 shows an instantaneous image at $t$ = 119ms, which is during the stable phase of the drug spray at actuation g (see Fig 1c). Three distinct regions were identified. Region (i) (surface waves) is the continuous jet liquid sheet. Although, the interaction between the air and liquid sheet is not well understood, it is assumed that an aerodynamic instability in the form of Kelvin-Helmholtz waves grows on the liquid sheet and causes the sheet to break up into horizontal ligaments, illustrated in region (ii). The ratio between the disrupting air forces and surface tension forces of the liquid sheet is expressed in terms of the gas weber number ($We_g = \rho U^2 l/\sigma$) which was calculated to be 1.06 to 2.71, where $\rho$ is the gas density, $U$ is the average spray velocity (16.5m/s to 26.25m/s), $l$ is the characteristic length taken as the orifice diameter of 0.28mm [33] and $\sigma$ is the liquid surface tension. For nasal spray applications, $We_g < 2$ suggesting long wave instabilities in the form of sinuous waves are dominant [34, 35]. These instabilities produce fluctuations in velocity and pressure which deforms the liquid sheet into ligaments.

The ligaments are identified as liquid threads which extend and stretch from the continuous liquid sheet. They form at the sheet break-up length (or distance from the nozzle) and it is evident that this length fluctuates over time. The ligaments continue to break up into droplets

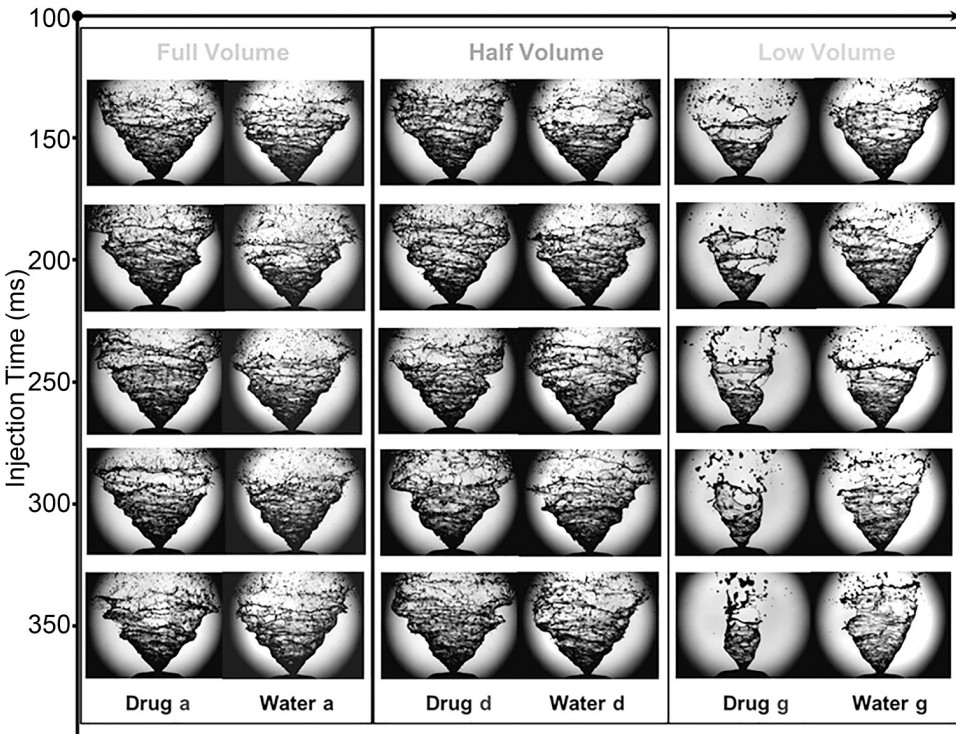

**Fig 7. Spray at different time instances (150ms, 200ms, 250ms, 300ms and 350ms) during early, mid and late actuations which represents full, half and low bottle volume respectively.**

due to the actuation pressure, ambient condition, nozzle geometry and liquid properties. Fig 8c demonstrates this process where corrugated ligaments break up into smaller droplets. The ligament size was approximately in the range of 90 − 500m (smallest during ligament necking).

Fig 9 shows the development of ligament formation. Initially, the continuous liquid sheet is deformed which results in thinning of its structure until holes develop in the liquid sheet. These holes become elongated prior to the formation of the thin corrugated ligaments (in Fig 8c). Eventually, the thin ligaments break into small droplets. In Fig 9a), three holes extend and stretch to form a ligament whereas in Fig 9b, a single hole is found.

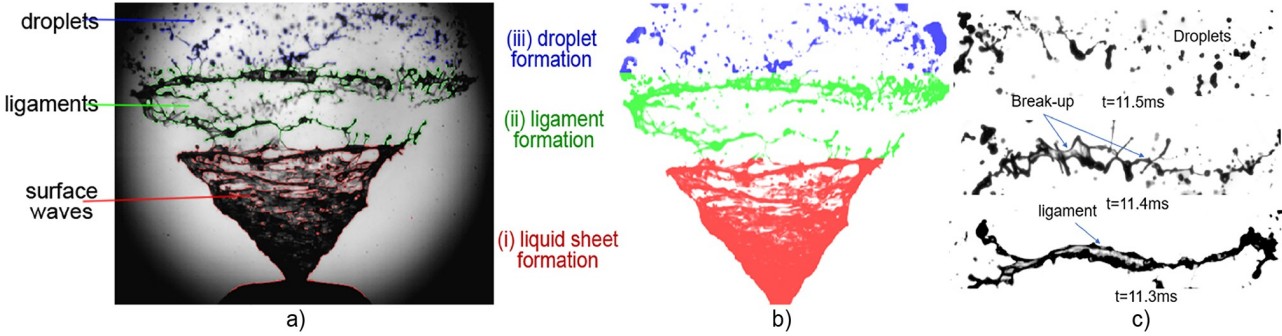

**Fig 8. a) Raw image b) Postprocessed image and c) Transition of liquid ligament to droplets.** The spray formation is labelled by (i) Sheet formation (ii) Ligament formation and (iii) Droplet formation identified by different colors red, green and blue respectively.

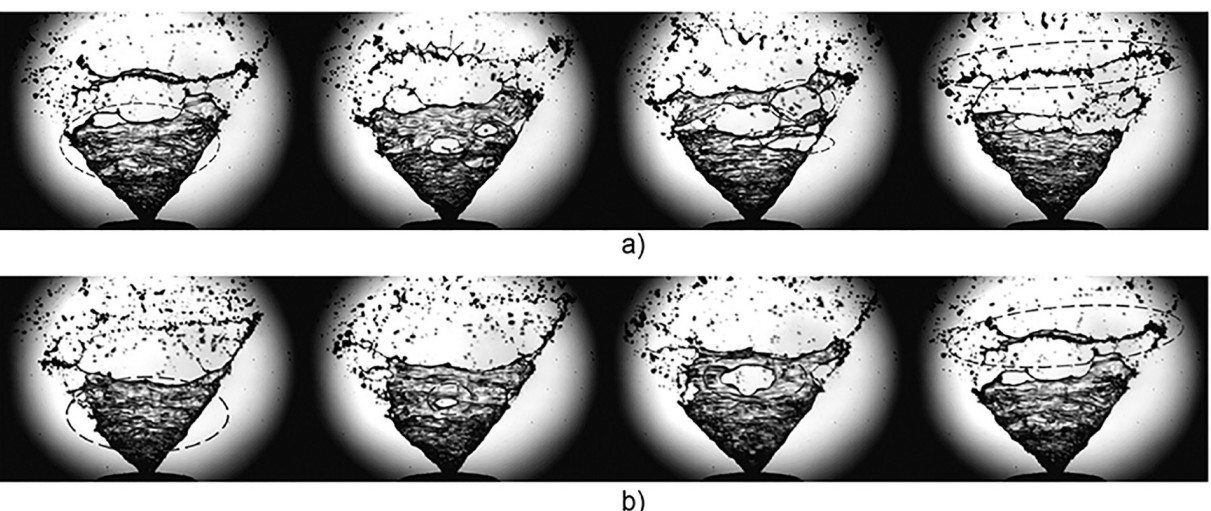

**Fig 9. Multi and single hole sheet breakup to form ligaments.**

Nasal spray characteristics does not solely depend on nozzle design and internal nozzle flow. The liquid properties of the ejected fluid also influence spray characteristics. The addition of excipients such as buffers, solubilizers, preservatives, surfactants, bio-adhesive polymers and penetration enhancers lead to changes in the liquid properties of the drug formulation. Polymers such as methylcellulose (MC) and hydroxypropyl methylcellulose (HPMC) or polyacrylic acid derivatives act as bio-adhesives and enhance viscosity. Increase in liquid viscosity increases the break-up length as highly viscous fluid impedes liquid disintegration [32]. This phenomenon was identified in the current study when the drug formulation, which was comparatively more viscous compared to water, had a somewhat longer break-up length (longer by ≈ 0.5mm).

The formulation in the nasal spray used contains oxymetazoline hydrochloride 0.05% and exhibits thixotropic behaviour due to its viscosity-enhanced formulations. Its rheologic behaviour is demonstrated by its viscosity at resting state which is high (3761cP [24]) but, under shear forces, its viscosity reduces, and hence the requirement that some nasal sprays must be shaken to reduce viscosity so the formula can easily pass through the atomizer. During spray atomization, the formulation stretches and swirls as it exits the nozzle orifice, thinning out and shearing into ligaments and later, into droplets. However, after depositing on the nasal mucosa, the formulation increases in viscosity in order to enhance its residence time on the surface and create the 'no drip' effect.

A lower surface tension liquid has higher tendency to disintegrate. The drug formulation has a lower surface tension than water. However, the effect of formulation on sprays can be described with the Ohnesorge number ($Oh$) which characterizes the effect of viscous to inertial and surface forces. Comparison of Oh numbers for the drug formulation and water indicate a greater influence of viscosity ($Oh_{drug} = 2-12Oh_{water}$) with the assumption that inertial force is the same ($\Delta P = 5$bar actuation force). The liquid properties also influence secondary break-up as they affect the liquid core length and ligament diameter. An increase in formulation viscosity (with similar surface tension) is expected to reduce plume angle and produce both larger and more variable droplets irrespective of the nasal spray devices tested. However, studies have shown that plume angles were identical and had no influence on droplet size distribution

when fluid with varying surface tension was used (with similar viscosity) and irrespective of the nasal devices tested [7, 19, 29].

## Implementation in CFD modelling

Spray modelling has been well understood and validated for high-pressure applications such as industrial and combustion fuel sprays [36, 37]. However, there are limited studies with accurate validation of low-pressure applications such as nasal sprays. The available primary break-up models, such as Huh and LISA [9, 38] were modelled for combustion sprays operating under high injection and combustion chamber pressure.

The current study aims to provide insight into the external and near-nozzle spray characteristics of a low-pressure nasal spray atomizing in atmospheric pressure. The high speed imaging was processed to obtain quantitative data to serve as a reference for initial conditions required for computational atomizer breakup models, such as the LISA model and an alternative approach involving explicitly defining the droplet parcels at the disintegration/break-up length (liquid core length) based on our measurements from a nasal spray. Secondary break-up would be achieved through the Taylor-Analogy-Breakup (TAB) model which is suitable for low Weber number applications.

The LISA breakup model requires a spray cone and dispersion angle as inputs. The spray cone angle describes the spray plume development whereas the dispersion angle describes the liquid sheet fluctuation from the mean cone spray angle. The dispersion angle is an important parameter in the LISA break-up model, as it leads to the radial droplet dispersion from the mean cone spray angle. This parameter has not been reported for nasal spray applications and past studies have used a dispersion angle of 3˚ [6]by tuning the parameter to match a droplet size distribution. For engine sprays, the dispersion angle was generally set to 10˚ [10, 11]. Our measured results showed an angle of 8.65˚±0.64˚.

An alternative to atomizer breakup models is to explicitly define the initial droplet conditions, which includes a choice of hollow-cone, ring cone and solid-cone, as well as a custom user-defined cone. The most likely choice of cone to best represent pressure-swirl atomizers would be the hollow cone model [33]. In this approach the primary break-up (eg: LISA, Huh models) is not modelled, but rather the droplet conditions in the near-nozzle are defined. This information can be extracted from measurements, that includes droplet location at the break-up length where the liquid sheet disintegration occurs, and spray cone angle. The droplets are distributed on a hollow circular ring at a break-up length from the nozzle, and a droplet size distribution would be imposed based on an empirical Rosin Rammler distribution function within the nasal spray droplet size distribution range.

A fully-resolved model of spray atomization is computationally intensive and challenging. Our measurement data of near-nozzle characteristics includes spray cone angle, break-up length, ligament diameter (break-up diameter), and dispersion angle and aims to contribute to the existing dataset for spray atomization CFD model setup.

## Conclusion

Liquid discharged from a nasal spray device was captured with a high speed and high-resolution camera to visualise the near nozzle characteristics of a transient spray and determine realistic values required to initialize spray primary breakup models. Different stages of the spray development of water and drug were identified using image processing techniques. The liquid jet sheet edges were superimposed to obtain essential information about the near nozzle spray characteristics such as break-up length (mean = 4.56mm), cone angle (mean = 29.68˚), dispersion angle (mean = 8.65˚)and break-up radius (mean = 4.85mm) which are pertinent to CFD

sub-models for spray atomization. Near-nozzle characteristics of the nasal spray were compared for different bottle volumes representing full-volume, half-volume and low-volume for both distilled water and a drug formulation. The liquid sheet was highly unstable due to its oscillating nature and it was observed that due to the lower viscosity, the spray from water retained its maximum cone angle earlier compared to the drug formulation. Thinning of the liquid sheet occurred prior to the formation of holes in the sheet. These holes further extended and stretched to form a thin corrugated ligament which eventually broke down to form droplets due to surface tension forces. These results identified parameters which are useful to determine characteristics of the spray evolution and are valuable information to be used as initial conditions for CFD modelling.

## Author Contributions

**Conceptualization:** Narinder Singh, Kiao Inthavong.

**Funding acquisition:** Kiao Inthavong.

**Investigation:** Kendra Shrestha, Narinder Singh, Kiao Inthavong.

**Methodology:** James Van Strien.

**Project administration:** Kiao Inthavong.

**Resources:** James Van Strien.

**Supervision:** Kiao Inthavong.

**Visualization:** James Van Strien.

**Writing – original draft:** Kendra Shrestha.

**Writing – review & editing:** Narinder Singh, Kiao Inthavong.

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
