## [Decision Letter · Decision Letter 0]

13 May 2020

PONE-D-20-09687

Primary break-up and atomization characteristics of a nasal spray

PLOS ONE

Dear Dr. Inthavong,

Thank you for submitting your manuscript to PLOS ONE. After careful consideration, we feel that it has merit but does not fully meet PLOS ONE’s publication criteria as it currently stands. Therefore, we invite you to submit a revised version of the manuscript that addresses the points raised during the review process.

We would appreciate receiving your revised manuscript by Jun 27 2020 11:59PM. To enhance the reproducibility of your results, we recommend that if applicable you deposit your laboratory protocols in protocols.io, where a protocol can be assigned its own identifier (DOI) such that it can be cited independently in the future. For instructions see: http://journals.plos.org/plosone/s/submission-guidelines#loc-laboratory-protocols

We look forward to receiving your revised manuscript.

Kind regards,

Josué Sznitman

Academic Editor

PLOS ONE

Journal Requirements:

Reviewers' comments:

Reviewer's Responses to Questions

**Comments to the Author**

1. Is the manuscript technically sound, and do the data support the conclusions?

Reviewer #1: Yes

Reviewer #2: Partly

2. Has the statistical analysis been performed appropriately and rigorously? 

Reviewer #1: Yes

Reviewer #2: Yes

3. Have the authors made all data underlying the findings in their manuscript fully available?

Reviewer #1: Yes

Reviewer #2: Yes

4. Is the manuscript presented in an intelligible fashion and written in standard English?

Reviewer #1: Yes

Reviewer #2: No

5. Review Comments to the Author

Reviewer #1: The authors presented findings describing nasal spray atomization break up with close-up visualizations of the break up process. These results are important to advance current knowledge since they will help in developing realistic nasal spray models for topical nasal drug delivery. The manuscript was detailed and well written.

Reviewer #2: The article “Primary break-up and atomization characteristics of a nasal spray” presents interesting visual results regarding the formation of a spray cone from a nasal spray device. These results are of importance in the field of nasal spray formulation and deposition since the spray characteristics are critical to the resulting droplet impaction within the nasal cavity. The use of several advanced data treatments (detailed image analysis, Canny edge detection, a brief introduction using the Weber number to describe air-surface tension relationships) is included in the manuscript, but insufficient information regarding how these can be/will be used to parameterize the proposed LISA model is described. The reviewer is left to question whether a follow-up publication reporting on the development of the LISA model to describe these results is planned. The limited quantitative evaluation of the results or their application to a broader set of nasal sprays provided in the manuscript is a significant weakness.

The authors conducted a somewhat limited study regarding spray formation using a single spray device containing a commercially-available formulation or water sprayed from the same device. The report would be strengthened by the inclusion of specific information about the spray actuator system used in the product tested (manufacturer, model, any performance or design specifications available) and about the properties of the specific formulation tested. The authors, instead, rely on the description of a range of formulation variables obtained from the literature (Table 1). This results in the inability to build upon and potentially generalize beyond the specific results provided.

The authors provide results obtained at different fill levels in the spray bottle. The interpretation of these results would be enhanced if the length of the dip tube and the dimensions of the spray bottle were included or, preferably, if a relationship between the height of the fluid in the reservoir at each fill level relative to the depth of the dip tube in the fluid volume was provided.

Inadequate discussion of the effects of surface tension and viscosity are provided. Very general, qualitative descriptions of the effect of these variables (e.g. “…holds the fluid together and resists disruptions and instabilities…” (line 173-4)) are included while a mechanistic understanding of the effects of these variables are available in the literature and should be provided in summary form in the discussion in this manuscript.

The manuscript needs careful copy editing prior to the next submission. There are numerous typos, grammatical errors and generally careless mistakes in figure descriptors. The references are also poorly edited, especially with respect to consistency in journal names.

6. PLOS authors have the option to publish the peer review history of their article (what does this mean?). If published, this will include your full peer review and any attached files.

Reviewer #1: No

Reviewer #2: No

---

## [Author Response · Author response to Decision Letter 0]

23 Jun 2020

Response to reviewer comments are in attachment

---

## [Editor Report · Decision Letter 1]

29 Jun 2020

Primary break-up and atomization characteristics of a nasal spray

PONE-D-20-09687R1

Dear Dr. Inthavong,

We’re pleased to inform you that your manuscript has been judged scientifically suitable for publication and will be formally accepted for publication once it meets all outstanding technical requirements.

Kind regards,

Josué Sznitman

Academic Editor

PLOS ONE
---

## [Editor Report · Acceptance letter]

23 Jul 2020

PONE-D-20-09687R1 

Primary break-up and atomization characteristics of a nasal spray 

Dear Dr. Inthavong:

I'm pleased to inform you that your manuscript has been deemed suitable for publication in PLOS ONE. Congratulations! Your manuscript is now with our production department. 

Kind regards, 

on behalf of

Prof. Josué Sznitman 

Academic Editor

PLOS ONE